# Longitudinal Associations of Dietary Sugars and Glycaemic Index with Indices of Glucose Metabolism and Body Fatness during 3-Year Weight Loss Maintenance: A PREVIEW Sub-Study

**DOI:** 10.3390/nu15092083

**Published:** 2023-04-26

**Authors:** Karen Della Corte, Elli Jalo, Niina E. Kaartinen, Liz Simpson, Moira A. Taylor, Roslyn Muirhead, Anne Raben, Ian A. Macdonald, Mikael Fogelholm, Jennie Brand-Miller

**Affiliations:** 1School of Life and Environmental Sciences and Charles Perkins Centre, University of Sydney, Sydney, NSW 2006, Australia; karendellacorte@gmail.com (K.D.C.); roslyn.muirhead@sydney.edu.au (R.M.); 2Department of Food and Nutrition, University of Helsinki, 00014 Helsinki, Finland; elli.jalo@helsinki.fi (E.J.);; 3Department of Public Health and Welfare, Finnish Institute for Health and Welfare, 00271 Helsinki, Finland; niina.kaartinen@thl.fi; 4Division of Physiology, Pharmacology and Neuroscience, School of Life Sciences, Queen’s Medical Centre, National Institute for Health Research (NIHR) Nottingham Biomedical Research Centre, Nottingham NG7 2RD, UK; liz.simpson@nottingham.ac.uk (L.S.); moira.taylor@nottingham.ac.uk (M.A.T.); 5Department of Nutrition, Exercise and Sports, Faculty of Science, University of Copenhagen, 1958 Copenhagen, Denmark; ara@nexs.ku.dk; 6Clinical Research, Copenhagen University Hospital–Steno Diabetes Center Copenhagen, 2730 Herlev, Denmark; 7Faculty of Medicine and Health Sciences, University of Nottingham, Nottingham NG7 2RD, UK

**Keywords:** dietary sugar, added sugar, glycaemic index, glucose metabolism, type 2 diabetes, overweight, body weight

## Abstract

Background: Dietary sugars are often linked to the development of overweight and type 2 diabetes (T2D) but inconsistencies remain. Objective: We investigated associations of added, free, and total sugars, and glycaemic index (GI) with indices of glucose metabolism (IGM) and indices of body fatness (IBF) during a 3-year weight loss maintenance intervention. Design: The PREVIEW (PREVention of diabetes through lifestyle Intervention and population studies in Europe and around the World) study was a randomised controlled trial designed to test the effects of four diet and physical activity interventions, after an 8-week weight-loss period, on the incidence of T2D. This secondary observational analysis included pooled data assessed at baseline (8), 26, 52, 104 and 156 weeks from 514 participants with overweight/obesity (age 25–70 year; BMI ≥ 25 kg⋅m^−2^) and with/without prediabetes in centres that provided data on added sugars (Sydney and Helsinki) or free sugars (Nottingham). Linear mixed models with repeated measures were applied for IBF (total body fat, BMI, waist circumference) and for IGM (fasting insulin, HbA1c, fasting glucose, C-peptide). Model A was adjusted for age and intervention centre and Model B additionally adjusted for energy, protein, fibre, and saturated fat. Results: Total sugars were inversely associated with fasting insulin and C-peptide in all centres, and free sugars were inversely associated with fasting glucose and HbA1c (Model B: all *p* < 0.05). Positive associations were observed between GI and IGM (Model B: fasting insulin, HbA1c, and C-peptide: (all *p* < 0.01), but not for added sugars. Added sugar was positively associated with body fat percentage and BMI, and GI was associated with waist circumference (Model B: all *p* < 0.01), while free sugars showed no associations (Model B: *p* > 0.05). Conclusions: Our findings suggest that added sugars and GI were independently associated with 3-y weight regain, but only GI was associated with 3-y changes in glucose metabolism in individuals at high risk of T2D.

## 1. Introduction

In parallel with obesity, type 2 diabetes (T2D) represents a major public health concern, with the incidence and/or prevalence of comorbidities continually increasing [1,2]. To date, modifiable risk factors for T2D include sugar-sweetened beverage (SSB) intake [3,4], but evidence for other sources of dietary sugars is scant [5]. The glycaemic index (GI), a measure of the postprandial impact of dietary carbohydrates, has a more consistent positive association with the risk of T2D than dietary sugars [6]. In contrast, dietary sugars (more specifically SSB), have been identified as key contributors to overweight and obesity [7,8]. Nonetheless, the development of dietary guidelines has been hindered by inconsistent findings arising from differences in amounts, sources, and forms of sugar as well as inadequate study design. Sugars consumed in liquid form may be less satiating than in solid forms [9], although total meal replacements in the form of liquid ‘shakes’ (50% energy as sugars) have been successfully used in rapid weight loss interventions. In *ad libitum* and excess energy settings, added sugars appear to promote weight gain but not when exchanged for starches or other carbohydrates [10,11,12].

Although T2D is a disease of abnormal carbohydrate metabolism, meta-analyses from prospective cohort studies have shown no clear association between the sum of all digestible carbohydrates and incident T2D [13,14,15]. In contrast, epidemiological findings support the concept that the structure (physical and chemical) of carbohydrate-rich foods rather than their quantity has the strongest effect on health outcomes [6,16,17]. Reducing postprandial glycaemia may be more relevant to lowering the risk for T2D by reducing beta-cell demand [18]. In observational studies, higher energy-adjusted GI and total glycaemic load (GL) were found to be independent risk factors for T2D [19]. In relatively short randomised controlled trials, lower GI diets resulted in improvements in glucose homeostasis in some studies [20,21], but not in others [22].

The PREVIEW (PREVention of diabetes through lifestyle Intervention and population studies in Europe and around the World) study was a long-term randomised controlled weight loss maintenance trial designed to test the effects of four diet and physical activity interventions in a 2 × 2 factorial design on the incidence of T2D in individuals with overweight or obesity and pre-diabetes after an initial 8-week weight loss period on a predominately liquid low-energy diet [23]. In post hoc analyses of PREVIEW, dietary GI and GL were positively associated with weight regain and deteriorating glycaemic status [24]. The aim of the present secondary analysis was to investigate longitudinal associations of added, free and total sugars intakes [25], as well as GI as a comparator, with indices of glucose metabolism (IGM) (fasting plasma glucose, insulin, C-peptide and HbA_1c_) and indices of body fat (IBF) (total body fat, waist circumference, and BMI) during a 3-year weight loss maintenance phase in those centres that provided data on added and/or free sugars.

## 2. Participants and Methods

### 2.1. Study Design

This study was a secondary observational analysis of the PREVIEW Study, a large 3-year, multinational, randomised intervention trial [23]. The original study was assessed in males and females aged 25–70 year, with overweight (BMI ≥ 25 kg/m^2^) and pre-diabetes. The study comprised a weight-loss phase followed by a weight-loss maintenance (WLM) phase [23]. Briefly, during the first 8-weeks, participants adhered to a low-energy diet (800 kcal/d) with the aim to lose ≥8% of initial body weight. Those who met the target and were T2D-free or had prediabetes after weight loss were eligible for the WLM phase. In this stage (duration 148 weeks), the effects of a healthy high-protein low-GI diet (HPLG) [protein 25% of energy intake (en%), carbohydrate 45 en%, GI ≤ 50] were compared with a healthy moderate-protein moderate-GI (MPMG) diet (protein 15 en%, carbohydrate 55 en%, GI ≥ 56) in combination with one of two exercise regimens (high-intensity or moderate-intensity exercise) [23]. The main results were published previously [26]. The current sub-study is an observational analysis based on data from the WLM phase (8 to 156 weeks), irrespective of original randomisation. The start of WLM (at 8 weeks) was considered the baseline for this analysis.

Only participants from the Sydney, Helsinki, and Nottingham cohorts of PREVIEW could be included because they had data on added or free sugars either from the national food composition tables or through calculation. Data were collected between August 2013 and March 2018. Overweight and obesity were defined as a BMI of 25.0–29.9 kg/m^2^ and ≥30.0 kg/m^2^, respectively. Prediabetes and T2D were evaluated in accordance with American Diabetes Association criteria [27].

Self-reported dietary intakes were assessed at 26, 52, 104, and 156 weeks. Dietary data at 26-weeks were used to estimate the diet at 8 weeks. Reported dietary intake was assessed using 4-day weighed dietary records which included four consecutive days including one weekend day. Participants were instructed how to use scales and conventional household measurements and to record in detail all foods and beverages consumed. Food records were reviewed by research dietitians during the clinical investigation days to assess the adequacy of the information. Research dietitians entered the individual food records by means of proprietary software in use in each centre (FoodWorks Professional, AivoDiet, Nutritics).

The GI of each food was obtained by use of GI databases. For mixed meals and some recipes, the weighted mean GI of the meal components was calculated [28] as previously described by Louie et al. [29]. Total GI and GL were calculated according to the formula of van Woudenbergh et al. [30]. Depending on national databases, added and free sugars were calculated within each intervention centre based on total and natural sugars amounts [29,31]. “Total sugars” refer to all monosaccharides (glucose, fructose, galactose) and disaccharides (sucrose, maltose, lactose), including lactose present in milk products and sugars contained within the cellular structure of foods (e.g., whole fruits). “Added sugars” were defined as sugars added to foods during processing, manufacturing, or home preparation (including honey, molasses, fruit juice concentrate, brown sugar, corn sweetener, sucrose, lactose, glucose, high-fructose corn syrup, malt syrup). “Free sugars” also included sugars naturally present in unsweetened fruit juices. The intervention centres of Sydney and Helsinki reported on added sugars and Nottingham reported on free sugars.

Information on age, sex, stature and ethnicity was collected with self-administered questionnaires at week 0. On every clinical investigation day, body weight was measured when participants were in a fasting state, with an empty bladder and wearing light clothing or underwear. BMI was calculated as body weight in kilograms divided by square height in meters. Fat mass was obtained using GE Lunar prodigy (eCORE 2005 software version 9.30.044) at Nottingham, Hologic Discovery (APEX software version 4) at Sydney and bioelectrical impedance (InBod 720 Body Composition Analyzer 2004; Biospace Co., Ltd., Seoul, Korea) at Helsinki. Waist circumference was measured when participants were at the end of breath expiration, at the midway point between the bottom of the rib cage and the top of the iliac crest. All staff were trained in the standard operating procedures at joint training sessions. All study participants provided written informed consent prior to commencing screening measurements. The study was approved at all intervention centres by the local human research ethics committees and was conducted in accordance with the latest revision of the Declaration of Helsinki (59th WMA General Assembly, Seoul, Republic of Korea, October 2008). University of Sydney: HREC approval on 24 June 2013, then subsequent approval by Sydney Local Health District Human Ethics Research Committee (SLHD HREC) on 8 April 2015 (Protocol No is X14-0408), which overrode the HREC approval. University of Helsinki: Coordinating Ethical Committee of HUS (Helsinki and Uusimaa Hospital District); 18 June 2013. University of Nottingham: UK National Research Ethics Service (NRES); 13/EM/0259; 5 July 2013.

The main outcomes of interest were fasting plasma glucose, insulin, and C-peptide concentrations, and glycated hemoglobin A_1c_ (HbA_1c_) measured at each time point. The secondary outcomes of interest were fat mass, waist circumference, and BMI. Information on outcomes of interest was assessed at 8, 26, 52, 104 and 156 weeks. Fasting (>10 h) blood samples were initially stored at −80 °C at the intervention sites and transported to the accredited laboratory (Finnish Institute for Health and Welfare, Helsinki, Finland) to be analysed with an Architect ci8200 integrated system (Abbott Laboratories, Abbott Park, IL, USA). 

### 2.2. Statistical Analysis

Characteristics of the study population are presented as mean ± SD or median (25th, 75th percentile) for continuous variables and as absolute (relative) frequencies for categorical variables and presented in Appendix A. 

Participants from the Universities of Sydney and Helsinki were merged into one group for assessment of longitudinal associations of added and total sugars, and GI with changes in markers of glycaemic status and body fatness during the WLM phase. Data from the University of Nottingham were analysed separately because data on free sugars, not added sugars, were provided. Data were not pooled for all three centres because direct comparisons between added/free sugars to total sugars and GI were required. To achieve normal distribution in outcome variables log_e_ or square root transformations were used. Dietary variables (total, added and free sugars, GI, protein, total fat, saturated fat, starch, and fibre intakes) were energy-adjusted by the residual method to provide a measure of these nutrient intakes uncorrelated with energy intake [32].

Prospective associations between dietary sugars intake (total sugars, added sugars, free sugars) or glycaemic index and risk markers of type 2 diabetes (fasting insulin, fasting glucose, HbA_1c_, C-peptide) and indices of body fatness (body fat percentage, waist circumference, BMI) were analysed by the use of linear mixed models with repeated measures. The initial regression model (model A) included the predictors of energy-adjusted sugars intake (total, free, added) or GI as well as age and intervention centre. Adjusted models (model B) were built on model A and were constructed by individual examination of potential influencing covariates and inclusion of those which substantially modified the predictor–outcome associations (≥10%) or significantly predicted the outcome. Potential confounding covariates considered were (1) other dietary factors [intakes of total protein, total fat, saturated fat, trans fat, starch, GI, fibre, energy], (2) anthropometric factors [fat mass (%), BMI, waist circumference, hip circumference, body weight], (3) other factors [age, sex, ethnicity (Caucasian, Asian, African, Arabic, Hispanic, or other)]. The final model B included model A (including age and intervention centre) and additionally adjusted for the intakes of energy, protein, fibre, and saturated fat as well as body fat as fixed effects for glucose metabolism outcomes. These models were additionally adjusted for time as a fixed effect and intervention centre and participant identifier as random effects and an autoregressive model was applied for the covariance matrix. To retain comparability of results, models were adjusted identically for variables of glucose metabolism and body fatness and the building of the models was conducted for the primary exposures. 

For easier interpretation, results from the linear mixed models with repeated measurements are presented as adjusted least-square means (95% CI) by tertiles of the respective predictor, while *p*-values stem from models with the predictors as continuous variables. Hence, the linear mixed models included the energy-adjusted predictors as grouped tertiles versus continuous variables in order for adjusted least-square means to be calculated and presented. The linear mixed models with repeated measurements included up to five time points per participant, thus allowing for variability over time to be accounted for in summarised results. Data were analysed under the assumption that missing data were missing at random. Both complete-case and available-case analyses were included in this study. A sensitivity analysis was conducted at 26 weeks, which excluded dropout participants whose last investigation visit was at 6 months (*n* = 20/20 of dropouts for added/free sugars groups). SAS statistical software package version 9.2 (SAS Institute Inc., Cary, NC, USA) was used for all statistical analyses. Statistical significance was set at a *p*-value < 0.05.

## 3. Results

A total of 709 subjects entered the WLM phase at the three sites. Of these, 514 had available GI, dietary sugars, and main outcome information; 132 from the University of Sydney, 211 from the University of Helsinki, and 171 from the University of Nottingham. Characteristics of the participants at baseline are presented in Appendix A. In the Sydney/Helsinki group (70.0% females), the median baseline BMI kg/m^2^ was 29.0 (range 26.1–33.1), and mean total and added sugars intakes (±SD) were 17.5% ± 4.6 and 4.2% ± 2.9 of total energy, respectively. For the Nottingham group (56.1% female), the median baseline BMI kg/m^2^ was 29.8 (range 27.2–33.6), and mean total and free sugars intakes (±SD) were 20.1% ± 6.7 and 5.6% ± 3.6 of total energy, respectively. Results for the sensitivity analyses produced similar findings as the main results. 

### 3.1. Indices of Glucose Metabolism

Added sugars intake was not associated with fasting insulin, fasting glucose, HbA_1c_ or C-peptide in the adjusted model B (all *p* > 0.05; see Figure 1 and Table 1). Free sugars intake was inversely associated with fasting plasma glucose (*p* = 0.008) and HbA_1c_ (*p* = 0.035) (see Figure 2 and Appendix A). Total sugars intake was inversely associated with fasting insulin, fasting glucose and C-peptide in the Sydney and Helsinki groups (Model B: all *p* < 0.05; see Figure 1, Table 1) and with fasting insulin and C-peptide in the Nottingham group (see Appendix A). The intake of fruit as a source of naturally-occurring sugar showed protective associations with IGM (see Appendix A). 

GI was positively associated with fasting insulin (*p* = 0.003), HbA_1c_ (*p* = <0.001), and C-peptide (*p* = 0.005) in adjusted Model B for Sydney and Helsinki (Figure 1 and Table 1). In the Nottingham group, GI was positively associated with fasting insulin (*p* = 0.015) (see Figure 2 and Appendix A). 

### 3.2. Indices of Body Fatness

Added sugars were positively associated with body fat percentage (*p* < 0.001) and BMI (*p* = 0.006). GI was positively associated with increases in waist circumference in the Sydney and Helsinki group (*p* < 0.001) (see Figure 1 and Figure 2) and in the Nottingham group (*p* = 0.019) (see Appendix A). In Sydney and Helsinki, total sugars intake was positively associated with body fat percentage (*p* = <0.001; Figure 1 and Table 2). 

## 4. Discussion

In this 3-year longitudinal analysis of individuals at high risk of T2D, higher added sugars intake predicted increasing body fat but was not associated with markers of glucose metabolism. In contrast, higher GI was associated with an increased waist circumference and glucose intolerance in the Sydney and Helsinki group. Interestingly, these trends were not seen with total sugars or free sugars intake, which were associated with improvements in glucose metabolism and were not linked with IBF. This may reflect other components in the food matrix or the type of sugar (glucose, fructose, sucrose, lactose) present.

Anecdotally, many health professionals assume that added sugars increase the risk of developing T2D. However, average amounts of added sugars intake have either no relationship to incident T2D [33,34,35], or even a beneficial association with insulin sensitivity and fasting insulin [36,37]. Indeed, a meta-analysis of 15 prospective cohort studies reported no association between total sugars intake and T2D, while higher sucrose consumption was associated with *decreased* risk [5]. High amounts of sugars in liquid form, including SSBs, have been positively associated with the risk of T2D [3,38,39,40,41,42,43]. However, epidemiological investigations of all added sugars intake often showed no association [33,35,36,38,44,45,46,47,48,49,50]. Sugars consumed in high amounts that exceed energy needs are likely to have different physiological effects to those consumed in energy balance [51,52].

In contrast to total carbohydrates and added sugars, many observational studies suggest that GI is a predictor of the development of T2D [15,53,54,55,56], although not all evidence is consistent [17,57]. Meta-analyses of RCTs have concluded that a low-GI diet is more effective than a high-GI diet in improving glucose metabolism [20,21]. However, to our knowledge, the PREVIEW study and this sub-study are the first to show an association between dietary GI and HbA_1c_ [24], a marker of average blood glucose concentration over the previous 8–12 weeks, which is now used to diagnose T2D when elevated. While the intake of carbohydrates may drive insulin resistance through a number of proposed mechanisms, saturated fatty acid intake has also been closely tied to insulin resistance development. In a supplementary analysis, we showed that saturated fatty acid intake was significantly associated with fasting insulin and C-peptide (see Appendix A). Thus, the GI is one dietary risk marker for developing insulin resistance that should be considered alongside saturated fat intake. 

In contrast to T2D, excessive consumption of sweetened drinks, and to a lesser extent sweetened foods, is widely acknowledged to have a role in the promotion of weight gain in prospective and experimental studies [58,59,60,61,62,63,64]. Nonetheless, the relevance of these findings to typical consumption habits has been questioned [65,66]. In a meta-analysis of controlled feeding trials, fructose intake produced weight gain only in settings where fructose was provided as excess energy, but not when isoenergetically exchanged for other carbohydrates [10]. A second meta-analysis of RCTs reported that isoenergetic exchange of starch for sugars had no effect on weight gain. Still, in cohort studies SSB led to significant weight gain in those with the highest vs lowest SSB intake, suggesting that this source may have unique effects [11]. If energy intake is not suppressed proportionately after the consumption of liquid sugars [67,68,69], targeting SSB as a source of excess energy is a prudent strategy. However, both dietary sugars and GI were associated with differing indices of body fat in the current study, implying that replacing sugars with sources of high-GI/GL starch may not be helpful. Other intervention studies have shown that consuming a low-GI diet assists with weight control compared to other conventional diets [70].

The beneficial association between *total* sugars (including free sugars) and glucose metabolism may be explained by the favourable effects of particular micronutrients and bioactive polyphenols found within natural sources of sugars and the fact that many of these foods have a lower GI (e.g., whole fruit, milk and dairy products). A high intake of fruits that are naturally high in fructose is associated with good metabolic health [66]. Further, when the main sources of dietary fructose are fruits and vegetables in their whole form and not as juice or smoothie, prospective studies have shown inverse associations with the risk of incident T2D [71,72]. Indeed, fruit intake was associated with improved glucose metabolism in our study (see Appendix A). Previous studies have reported inverse associations between simple sugars and GI in people with diabetes [73], but in the present study, the intake of added sugars was associated with a higher GI. Many foods that are sweetened with added sucrose have shown either no association (cakes and cookies) or a protective association (whole-grain cereals, fruit, yogurt, and even ice-cream) with T2D [74,75]. 

There are notable strengths of the present study. First, we provide new evidence of the associations between dietary sugars and health markers during longer-term weight loss maintenance, which is more likely to address a life-long issue, particularly for individuals with obesity and increased susceptibility to T2D. Few studies of weight loss maintenance and weight regain have been undertaken and rarely for as long as 3 years. Secondly, unlike some studies with fixed energy content, we determined the associations in a free-living context and an ad libitum diet. Thirdly, because diet and health markers were measured regularly over 3 years, we were able to leverage multiple, longitudinal, real-time data. This large dataset allowed adjustments for multiple confounders. We investigated the role of three types or sources of sugars (added, free and total) as well as multiple outcome indices for both glucose metabolism and body fatness after a period of rapid weight loss. Finally, the findings may be more generalizable because three different countries, albeit similar in terms of economy and food culture, were represented.

Several limitations should also be noted. The GI and dietary sugars intakes were calculated from self-reported weighed 4-day food diaries in individuals whose added sugars intake was relatively low (4.2% ± 2.9% of total energy compared to 9.0% ± 7.5% of total energy reported in the Australian population based on national survey data for this age group) [76]. Therefore, in interpreting these results it should be noted that the relatively low intake amounts of added and free sugars may limit the ability to observe in these data an association with markers of glycaemia. Although these records estimate food intake more accurately than food-frequency questionnaires, misreporting can occur [77]. It is possible that sugar-rich foods were selectively underreported due to their perceived unhealthiness. Further, low GI may be a proxy for a certain type of diet, including one which is rich in fruit, vegetables, legumes, berries, and dairy foods. Although we have tested and adjusted for dietary macronutrient composition, there are several dietary components (e.g., vitamins, minerals, and polyphenols) that we could not adjust for, hence residual and unmeasured confounders may exist. Finally, our findings may not be relevant to individuals with normal weight and glucose tolerance.

In conclusion, this secondary analysis of a 3-year weight loss maintenance intervention found evidence that a higher intake of added sugars and a higher dietary GI was associated with increased body fat or waist circumference. In contrast, only GI was an independent predictor of worsening glucose status. Paradoxically, total and free sugar intakes were associated with improved glucose metabolism. Taken together, the findings from this particular study population add to the increasing body of evidence that the glycaemic index of foods and hence fluctuations in postprandial glycaemia may be more associated with increased future risk of T2D than added or free sugars intake. 

## Figures and Tables

**Figure 1 nutrients-15-02083-f001:**
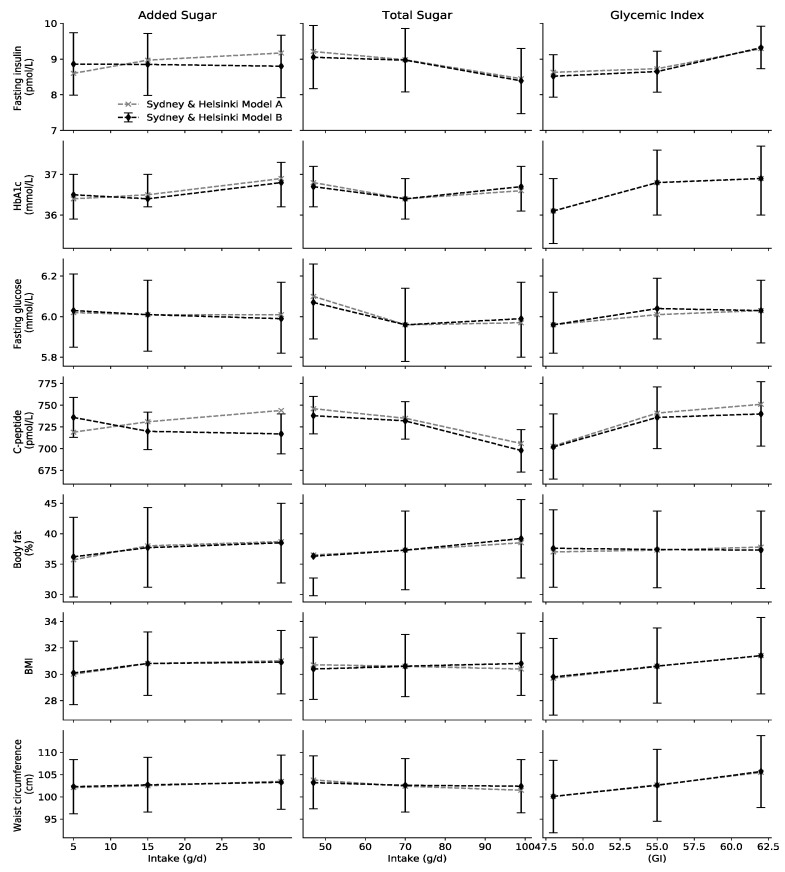
Plasma levels of fasting insulin, glucose, C-peptide, and HbA1c (glycated hemoglobin), as well as body fat, BMI (body mass index) and waist circumference by tertiles of added sugar, total sugar and glycaemic index for the combined Sydney and Helsinki intervention groups (*n* = 343). Data are generic least square means and 95% CI for model A (adjusted for age and intervention centre) and model B (additionally adjusted for intakes of energy, protein, fibre, and saturated fat). Indices of glucose metabolism were also adjusted for body fat.

**Figure 2 nutrients-15-02083-f002:**
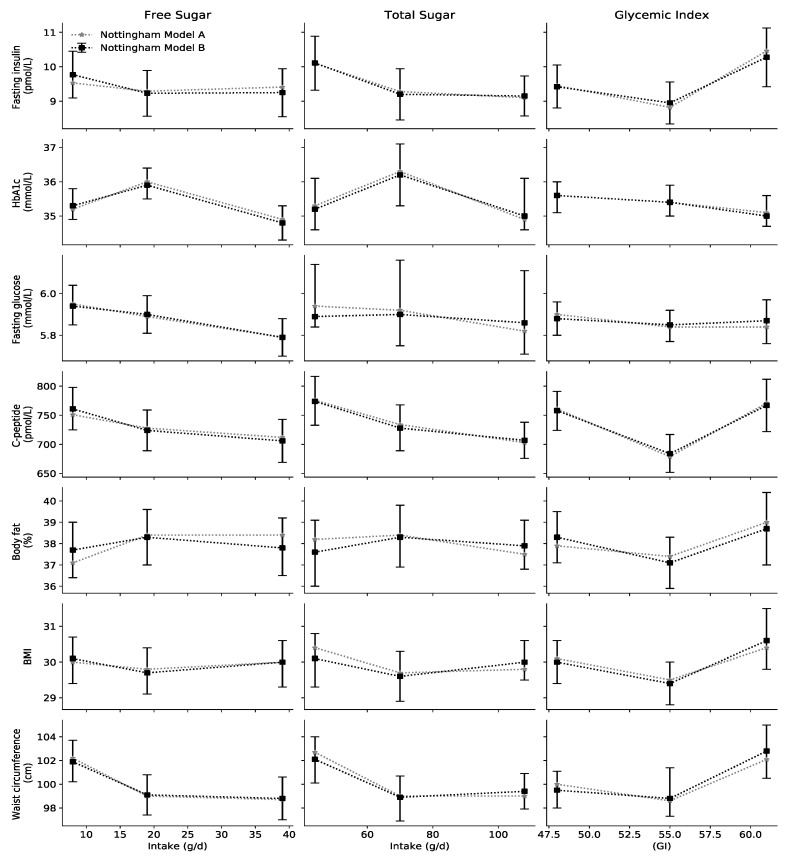
Plasma levels of fasting insulin, glucose, C-peptide, and HbA1c (glycated hemoglobin) as well as body fat, BMI (body mass index) and waist circumference by tertiles of added sugar, total sugar and glycaemic index for the Nottingham intervention group (*n* = 171). Data are generic least square means and 95% CI for model A (adjusted for age and intervention centre) and model B (additionally adjusted for intakes of energy, protein, fibre, and saturated fat). Indices of glucose metabolism were also adjusted for body fat.

**Table 1 nutrients-15-02083-t001:** Associations of added sugar, total sugar, and glycemic index with indices of glucose metabolism for Sydney and Helsinki intervention groups (*n* = 343).

	Tertiles of Added Sugar				Tertiles of Total Sugar				Tertiles of Glycemic Index			
	Low (T1)	Moderate (T2)	High (T3)	*P_trend_*	Low (T1)	Moderate (T2)	High (T3)	*P_trend_*	Low (T1)	Moderate (T2)	High (T3)	*P_trend_*
Intake (g/d) or GI ^a^	5 (4; 6)	15 (14; 16)	33 (33; 35)		47 (46; 48)	70 (69; 71)	99 (97; 100)		48 (46; 49)	55 (53; 56)	62 (60; 63)	
**Fasting insulin (pmol/L)**												
Model A	8.60 (8.20; 9.01)	8.97 (8.57; 9.38)	9.17 (8.77; 9.58)	**0.002**	9.21 (8.83; 9.61)	8.98 (8.59; 9.37)	8.45 (8.00; 8.89)	**0.009**	8.63 (8.19; 9.10)	8.73 (8.29; 9.17)	9.29 (8.87; 9.70)	**0.010**
Model B	8.86 (7.99; 9.74)	8.85 (7.98; 9.72)	8.80 (7.92; 9.67)	0.746	9.05 (8.17; 9.94)	8.97 (8.08; 9.86)	8.39 (7.47; 9.30)	**0.046**	8.52 (7.93; 9.12)	8.65 (8.07; 9.22)	9.32 (8.73; 9.92)	**0.003**
**HbA1c (mmol/mol)**												
Model A	36.4 (35.7; 37.1)	36.5 (35.8; 37.1)	36.9 (36.2; 37.6)	**0.019**	36.8 (36.2; 37.4)	36.4 (35.8; 37.0)	36.6 (36.0; 37.2)	0.155	36.1 (35.1; 37.1)	36.8 (35.8; 37.8)	36.9 (35.9; 37.9)	**<0.001**
Model B	36.5 (35.9; 37.0)	36.4 (36.2; 37.0)	36.8 (36.2; 37.3)	0.218	36.7 (36.2; 37.2)	36.4 (35.9; 36.9)	36.7 (36.1; 37.2)	0.566	36.1 (35.3; 36.9)	36.8 (36.0; 37.6)	36.9 (36.0; 37.7)	**<0.001**
**Fasting glucose (mmol/L)**												
Model A	6.02 (5.87; 6.18)	6.01 (5.86; 6.16)	6.01 (5.86; 6.16)	0.954	6.10 (5.94; 6.25)	5.96 (5.81; 6.11)	5.97 (5.82; 6.12)	**<0.001**	5.96 (5.84; 6.09)	6.01 (5.93; 6.18)	6.03 (5.90; 6.15)	0.209
Model B	6.03 (5.85; 6.21)	6.01 (5.83; 6.18)	5.99 (5.82; 6.17)	0.527	6.07 (5.89; 6.26)	5.96 (5.78; 6.14)	5.99 (5.80; 6.17)	**0.024**	5.96 (5.82; 6.12)	6.04 (5.89; 6.19)	6.03 (5.87; 6.18)	0.131
**C-peptide (pmol/L)**												
Model A	719 (650; 788)	731 (663; 801)	744 (674; 813)	**0.001**	746 (681; 811)	735 (670; 800)	706 (640; 772)	**0.038**	703 (615; 791)	741 (654; 829)	751 (663; 838)	**0.006**
Model B	736 (713; 759)	720 (699; 742)	717 (694; 740)	0.909	738 (717; 760)	732 (711; 754)	698 (673; 722)	**0.045**	702 (665; 740)	736 (700; 771)	740 (703; 777)	**0.005**

Values are adjusted least-square means (95% CIs) unless otherwise indicated. Linear trends (*P_trend_*) were obtained using a linear mixed model with repeated measures. The predictor of glycemic index as well as the transformed and energy-adjusted predictors of dietary added sugar intake and total sugar intake were used as continuous variables. Model A adjusted for age at time of study begin and intervention center. Model B additionally adjusted for body fat percentage, energy intake, protein intake, fibre intake and saturated fat intake. Transformations of variables for analysis: log_e_ for protein intake, saturated fat intake, energy intake, HbA1c, insulin and C-peptide; square root for total and added sugar intakes. HbA1c: glycated hemoglobin A1c. ^a^ Values are unadjusted medians (25th, 75th percentile). *p*-values stem from models with predictors as continuous variables. Bold values indicate significant findings (*p* < 0.05).

**Table 2 nutrients-15-02083-t002:** Associations of added sugar, total sugar, and glycemic index with indices of body fatness for Sydney and Helsinki intervention groups (*n* = 343).

	Tertiles of Added Sugar				Tertiles of Total Sugar				Tertiles of Glycemic Index			
	Low (T1)	Moderate (T2)	High (T3)	*P_trend_*	Low (T1)	Moderate (T2)	High (T3)	*P_trend_*	Low (T1)	Moderate (T2)	High (T3)	*P_trend_*
Intake (g/d) or GI ^a^	5 (4; 6)	15 (14; 16)	33 (33; 35)		47 (46; 48)	70 (69; 71)	99 (98; 100)		48 (46; 49)	55 (53; 56)	62 (60; 63)	
**Body fat (%)**												
Model A	35.7 (29.2; 42.1)	38.0 (31.6; 44.4)	38.7 (32.3; 45.1)	**<0.001**	36.5 (30.7; 42.4)	37.3 (31.5; 43.2)	38.5 (32.7; 44.4)	**<0.001**	37.0 (30.8; 43.2)	37.3 (31.1; 43.5)	37.8 (31.6; 44.0)	0.568
Model B	36.2 (29.6; 42.7)	37.7 (31.2; 44.3)	38.5 (31.9; 45.0)	**<0.001**	36.3 (29.8; 42.7)	37.3 (30.8; 43.7)	39.2 (32.7; 45.6)	**<0.001**	37.6 (31.2; 43.9)	37.4 (31.1; 43.7)	37.3 (31.0; 43.7)	0.656
**BMI**												
Model A	30.0 (27.8; 32.2)	30.8 (28.6; 33.0)	31.0 (28.8; 33.2)	**<0.0001**	30.7 (28.7; 32.7)	30.6 (28.6; 32.6)	30.4 (28.4; 32.3)	0.278	29.7 (27.0; 32.5)	30.6 (27.9; 33.4)	31.4 (28.7; 34.1)	**0.022**
Model B	30.1 (27.7; 32.5)	30.8 (28.4; 33.2)	30.9 (28.5; 33.3)	**0.006**	30.4 (28.1; 32.8)	30.6 (28.3; 33.0)	30.8 (28.4; 33.1)	0.233	29.8 (26.9; 32.7)	30.6 (27.8; 33.5)	31.4 (28.5; 34.3)	0.142
**Waist circumference (cm)**												
Model A	102.1 (96.6; 107.5)	102.5 (97.0; 107.9)	103.5 (98.1; 108.9)	**0.024**	103.8 (98.8; 109.0)	102.4 (97.3; 107.5)	101.5 (96.4; 106.6)	**<0.001**	100.1 (92.3; 107.8)	102.7 (94.9; 110.4)	105.5 (97.8; 113.3)	**<0.0001**
Model B	102.3 (96.2; 108.4)	102.7 (96.6; 108.9)	103.3 (97.2; 109.4)	0.188	103.2 (97.3; 109.2)	102.6 (96.6; 108.6)	102.4 (96.4; 108.4)	0.220	100.1 (91.9; 108.2)	102.6 (94.5; 110.7)	105.7 (97.6; 113.8)	**<0.0001**

Values are adjusted least-square means (95% CIs) unless otherwise indicated. Linear trends (*P_trend_*) were obtained using a linear mixed model with repeated measures. The predictor of glycemic index as well as the transformed and energy-adjusted predictors of dietary added sugar intake and total sugar intake were used as continuous variables. Model A adjusted for age at time of study begin and intervention center. Model B additionally adjusted for energy intake, protein intake, fibre intake and saturated fat intake. Transformations of variables for analysis: log_e_ for protein intake, saturated fat intake, energy intake, and BMI; square root for total and added sugar intakes. BMI: body mass index. ^a^ Values are unadjusted medians (25th, 75th percentile). *p*-values stem from models with predictors as continuous variables. Bold values indicate significant findings (*p* < 0.05).

## Data Availability

Data described in the manuscript, code book, and analytic code will be made available upon request pending application and approval of the trial steering committee.

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
