# Peer review of "Longitudinal Associations of Dietary Sugars and Glycaemic Index with Indices of Glucose Metabolism and Body Fatness during 3-Year Weight Loss Maintenance: A PREVIEW Sub-Study"

_nutrients, 2023, doi:10.3390/nu15092083_

Round 1

Reviewer 1 Report

Authors present an interesting, well-constructed paper examining the association between sugars, glycemic index and markers of type 2 diabetes.

I have 2 major comments related to the message and scope of the paper.

1. Food is understandably a package deal, thus, to discuss “sugars” independent of the food is somewhat disingenuous. For example, fruit intake is associated with reduced T2DM-associated clinical markers, yet, it is also high in sugars, while not “added”. This distinction is important to make, especially when discussing total sugars which may explain why there was no impact on total sugar intake and insulin. An analysis on the intake of whole fruit-derived sugars with the markers of interest may yield important distinctive findings.

2. Additionally, while added and refined sugars are important in the context of weight gain which then promotes insulin resistance due to increased fatty acid synthesis and oversupply, and also in the context of those who are diabetic and avoiding insulin and blood glucose spikes, to look at sugars alone in the context of insulin resistance development is misleading. Improper glucose handling is a symptom of the underlying insulin resistance, and carbohydrate consumption in itself does not drive insulin resistance. Indeed, fatty acid oversupply, aka lipotoxicity, is the primary driver of insulin resistance (see PMID: 24373240 for an in-depth review) which is closely tied to saturated fatty acid intake and adiposity independently. As such, I request that authors assess in parallel saturated fatty acid intake in this cohort of subjects, since this is a dietary nutrient most closely tied to insulin resistance development.

My minor comment is the following:

Regarding the statement in the methods: “healthy high-protein low-GI diet (HPLG) [protein 25% of energy intake (en%), carbohydrate 45 en%, GI ≤ 50] were compared with a healthy moderate-protein moderate-GI (MPMG) diet (protein 15 en%, carbohydrate 55 en%, GI ≥ 56)” What characterizes this diet as healthy? Was this considered a high "animal" protein diet? I am skeptical of such statements since “healthy, high protein” may be an oxymoron if animal food intake is high which typically parallels saturated fat intake.

Author Response

Response to comment 1:

We agree with the reviewer that the lack of associations observed between total sugar and T2DM-outcomes may be due to the favorable effects of healthier sources of sugar such as fruit. We also agree that food sources are important to consider with regards to sugar categories. Thus, in response to this request we have additionally included fruit intake as a predictor and included it in the supplementary material (please see the newly created Supplementary Table 3) for T2DM outcomes. You may notice that we have fruit intake in grams/day and servings/day based on the available data in the different intervention centres for this variable.

This additional information on fruit intake was noted in the results section (lines 188-189) and the discussion section (lines 251-253).

Response to comment 2:

The reviewer is correct that there are other important dietary predictors that may be involved with developing insulin resistance (IR). High intakes of dietary sugar and high-GI foods may lead to increased postprandial oxidation and the formation of advanced glycated end-products which may in turn damage cells and tissues and promote insulin resistance. Dietary sugars have also been shown to drive fatty acid synthesis, which may lead to increased ectopic fat and impaired insulin signaling. Thus the scope of this project was to investigate dietary sugar and glycemic index and their unique contribution to T2DM risk. However, we agree that saturated fatty acid intake is another important contributor, albeit somewhat out of the scope of this paper. Nonetheless, in line with this request we investigated the intake of saturated fatty acids as an additional predictor and included the results alongside those of fruit intake in Supplementary Table 3.

We lengthened the discussion by five lines and included this information on saturated fatty acid as a dietary determinant that should be considered alongside GI when assessing how diet affects insulin resistance (lines 225-230).

Response to comment 3:

The reviewer is correct that the source of protein (animal vs plant) needs to be considered when assuming that a higher protein diet is ‘healthy’. As outlined in more detail in PMC5490611, the food items with increased use in the higher protein group compared to the moderate protein group were poultry, fish, legumes, low-fat dairy products, pasta and whole grain cereals. We now include the citation after this sentence (line 76) to direct readers to the paper that describes the design and methods in more detail.

Reviewer 2 Report

This is a sub-study of the PREVIEW Study, a large 3-year, randomized intervention trial where the effect of the intake of different sugars (added sugar, total sugar) and the glycemic index on IBF and IGM variables, as surrogate variables for the development of type 2 diabetes mellitus, was studied. The study was carried out in 709 subjects after a period of weight loss. The authors found that intake of these sugars and the GI increase body fat and waist circumference, while added and free sugars improve glucose metabolism.

Concerns

The results obtained on the effect of simple sugars on insulinemia, basal glycemia, C peptide and A1c are striking (although for model B there are no differences in the IGM variables for added sugars). The authors justify these findings with references to other studies (ref. 3, 24-27), but ignore several meta-analyses that go in the opposite direction (Qin P et al Eur J Epidemiol. 2020 doi: 10.1007/s10654-020- 00655-y, McKeown NM et al. Diabetologia. 2018 doi: 10.1007/s00125-017-4475-0, Chen Z et al Diabetes Care. 2023 doi: 10.2337/dc22-1993). Authors should include these studies in the Discussion.

The study design is not clear to me. The manuscript indicates that the subjects were studied from week 26 to week 156, at least with regard to the analysis of the self-reported diets, but regarding the IBF and IGM variables, which week do they correspond to? the baseline or the last week of the study? Unless linear trends (Ptrend) have been used to analyze the evolution of the variables as a whole, assuming that there were many differences in the times that the subjects were studied. If this were not the case, the correct thing would be an analysis at the beginning and at the end of the intervention.

In this same line, the models developed were made based on the data at the beginning of the study (at least that is how it is specified for model A, and I assume that this criterion was also applied for model B), when in my opinion they should be done with the data of the end of the intervention. It is not in vain that it is alleged that this is a study whose strength is the follow-up time.

From the methodological point of view, I consider that it would be more correct to use the % of body fat and not the absolute values (in kg) of fat that are dependent on the weight of the subjects. I also believe that the HOMA index should have been included among the IGM variables, as it provides a better view of glucose metabolism.

Although the authors correctly point out that “Sugars consumed in high amounts that exceed energy needs are likely to have different physiological effects to those consumed in energy balance”, the truth is that adjusting the variables for energy intake (something that is usually done in epidemiological studies) is still an artifact (there are no poisons but doses), but the final message of this study is that the problem in relation to the risk of diabetes is the glycemic index and not so much the intake of added or free sugars. Something that I do not share since it does not correspond to real life. And this is something that should be mentioned more forcefully in the manuscript.

Author Response

Response to comment 1:

            Thank you for the additional meta-analyses on sugar-sweetened beverage intake (SSB) and T2D risk markers. There is indeed a large body of evidence establishing SSB intake (sugar in liquid form) and increased T2D risk. Our unique aim was to look at other sugar categories (total, free, added) and indices of glucose metabolism (IGM) besides SSB intake category that already has sufficient evidence on it. While we do compare our findings to other similar studies that show no association (or an inverse one) with IGM, we also point out that sugar in a liquid form (SSB) is detrimental and increases risk for T2D (see sentences 213-214). We unfortunately were not able to include SSB as a predictor because it was unavailable in our datasets.

According to this request, we included Qin et al and McKeown et al. in the discussion alongside the other meta-analyses on SSB and T2D (line 215).

Response to comment 2:

We used linear mixed models with repeated measurements. Each participant had up to 5 different time points for their dietary and outcome variables. Within such a model, a repeated statement for time could be inserted, thus allowing for variability overtime to be accounted for. As the reviewer correctly stated, we analyzed the evolution of the predictor/outcome relationship as a whole and the analysis was updated with every repeated measurement allowing for a summarized statistic to be reported.

In response to this comment, the methods section now includes a further explanation to help readers understand how these data were analyzed (see lines 165-167).

Response to comment 3:

Model A and Model B represent summarized findings of all 5 time points throughout the follow-up period. Model A is referred to as the base model because it does not include any of the identified confounders and simply shows the base relationship between the predictor and outcome. Model B is the adjusted model.

In response to this comment, we removed the word ‘base’ in relation to Model A because this could be conflated with ‘baseline’ measurement (line 145).

Response to comment 4:

We agree with the reviewer that body fat percentage would better reflect relative body fat than absolute values in kg. Thus, we performed a major revision and replaced the outcome of body fat (kg) with body fat (%), reran all analyses on this outcome and updated the tables, manuscripts, and figures accordingly.

Response to comment 5:

The reviewer is right that in this study population, the energy-adjusted GI is more relevant than energy-adjusted added sugar when it comes to T2D prevention. These findings challenge the common misconceptions regarding dietary sugar and T2D risk. We believe that the public focus on dietary sugar and T2D risk may cause more relevant findings relating to glycemic index and carbohydrate quality to be overlooked.

Round 2

Reviewer 1 Report

I thank and commend authors on addressing comments, the added information makes for a more compelling paper. Well done.

Author Response

Thank you.

Reviewer 2 Report

The authors have partially addressed my concerns. Although the argument of grouping the measurements to evaluate the different variables together is reasonable to determine the weight that the consumption of certain sugars has on adiposity and glucose metabolism indices, the fact that the duration of the study is considered a strength of the same, it seems necessary a temporal comparison between the beginning and the end of the study. After all, they are trying to find out about the effect of certain nutrients on the risk of developing T2DM. Although the authors concede that it is correct to use better body fat percentage, they have only done this for the Nottingham cohort. I have not seen any comments about my suggestion to use the HOMA index. Finally, in the conclusions, I consider that the last statement, regarding added sugars, does not correspond to the results obtained or in any case may lead to confusion. The fact that a higher glycemic index could increase the risk of T2DM does not imply that added sugars are harmless. Among other things because this is a controlled study and a high consumption of added sugars (among which SSB are included) in the general population is related to higher rates of obesity and therefore diabetes. These types of messages are dangerous, as they could be misinterpreted. For this reason, the authors should make it clear in their conclusions that these results come from a particular type of study and that they should not be extrapolated to the general population.

Author Response

Title of Paper:

Longitudinal associations of dietary sugars and glycemic index with indices of glucose metabolism and body fatness during 3-year weight loss maintenance: a PREVIEW sub-study

Authors:

Karen Della Corte, Elli Jalo, Niina E. Kaartinen, Liz Simpson, Moira A. Taylor, Roslyn Muirhead, Anne Raben, Ian A. Macdonald, Mikael Fogelholm, Jennie Brand-Miller

POINT-BY-POINT REPLY (ROUND 2)

EXPLANATION TO REVIEWER 2:

We have carefully read your further suggestions for change and thank you for your critical review.

All edits are marked in yellow in the manuscript.

Reviewer 2:

Comment 1:

The authors have partially addressed my concerns. Although the argument of grouping the measurements to evaluate the different variables together is reasonable to determine the weight that the consumption of certain sugars has on adiposity and glucose metabolism indices, the fact that the duration of the study is considered a strength of the same, it seems necessary a temporal comparison between the beginning and the end of the study. After all, they are trying to find out about the effect of certain nutrients on the risk of developing T2DM.

Response to comment 1:

In this study, we are not measuring the effect of any changes in the diet and comparing the findings at the study begin and end. This is a secondary observational analysis in which we pooled the data irrespective of original randomization. We were able to track changes over time and thereby gain a better picture of how the habitual diet relates to these outcomes, something that is more meaningful than cross-sectional snapshots at the beginning and end of the study. The linear mixed model with repeated measurements was applied in order to capture and summarize how these predictors (sugars and GI) relate to body weight and glucose metabolism over a three-year period.

Comment 2:

Although the authors concede that it is correct to use better body fat percentage, they have only done this for the Nottingham cohort.

Response to comment 2:

Please see in Table 1 and Figure 1 that analyses were rerun using body fat percentage for Sydney and Helsinki cohorts in addition to the Nottingham cohort.

Comment 3:

I have not seen any comments about my suggestion to use the HOMA index.

Response to comment 3:

We apologize for not directly responding to this point. We agree that the HOMA index would be especially useful as an outcome relating to insulin sensitivity/resistance. These variables are not available in the existing datasets and would need to be calculated. The original HOMA1 model contains a simple nonlinear equation that could easily be calculated but has been updated by the more sophisticated HOMA2 model. This calculation is involved and would require clinical judgement: e.g., exclusion of fasting glucose <2.5 mmol/l (hypoglycemic cases), etc. Thus, considering the limited time available to revise the manuscript it would not be possible for the authors to calculate, validate and include this outcome.

Comment 4:

Finally, in the conclusions, I consider that the last statement, regarding added sugars, does not correspond to the results obtained or in any case may lead to confusion. The fact that a higher glycemic index could increase the risk of T2DM does not imply that added sugars are harmless. Among other things because this is a controlled study and a high consumption of added sugars (among which SSB are included) in the general population is related to higher rates of obesity and therefore diabetes. These types of messages are dangerous, as they could be misinterpreted. For this reason, the authors should make it clear in their conclusions that these results come from a particular type of study and that they should not be extrapolated to the general population.

Response to comment 4:

We understand the reviewer’s concern that these findings run contrary to the general public view on sugar and diabetes risk. However, we also feel it our duty to report the results of the study as they are, without imposing our own preconceived biases relating to sugar. This is one of many studies to show that sugar consumed in moderate amounts in a non-liquid form is unrelated to T2D risk (as noted in the discussion and introduction). While at the same time, a larger body of evidence indicates that GI is more predictive. We also point out in this study that the source of sugar is important (fruit intake and thus total sugar showing a beneficial association). The main concluding sentence is not stating that sugar is harmless, rather that our study found that GI is more associated with future T2D risk than added or free sugars intake.

In response to this comment, we have added to the final sentence of the conclusion “the findings from this particular study population”(line 285).

We agree with the reviewer that added sugar could lead to weight gain which could lead to T2D. As stated in the conclusion, both added sugar and GI were associated with increased body fatness. Of note, we adjusted for body fat when assessing the relationship between sugar intake and indices of glucose metabolism.  
